# Characterization of 475 Novel, Putative Small RNAs (sRNAs) in Carbon-Starved *Salmonella enterica* Serovar Typhimurium

**DOI:** 10.3390/antibiotics10030305

**Published:** 2021-03-16

**Authors:** Dominika Houserova, Donovan J. Dahmer, Shivam V. Amin, Valeria M. King, Emmaline C. Barnhill, Mike E. Zambrano, Meghan A. Dean, Aline Crucello, Kevin M. Aria, Michael P. Spector, Glen M. Borchert

**Affiliations:** 1Department of Pharmacology, USA College of Medicine, Mobile, AL 36688-0002, USA; dh1001@jagmail.southalabama.edu (D.H.); djd1423@jagmail.southalabama.edu (D.J.D.); sva1002@jagmail.southalabama.edu (S.V.A.); kma1323@jagmail.southalabama.edu (K.M.A.); 2Department of Biomedical Sciences, University of South Alabama, Mobile, AL 36688-0002, USA; mspector@southalabama.edu; 3Department of Biology, University of South Alabama, Mobile, AL 36688-0002, USA; vmk902@berkeley.edu (V.M.K.); emmaline.walters@gmail.com (E.C.B.); mike.the.zambrano@gmail.com (M.E.Z.); dean.meghan@outlook.com (M.A.D.); alinecrucello@gmail.com (A.C.)

**Keywords:** carbon-starvation, noncoding RNA, *Salmonella*, sRNA, starvation-stress, transcriptomics

## Abstract

An increasingly apparent role of noncoding RNA (ncRNAs) is to coordinate gene expression during environmental stress. A mounting body of evidence implicates small RNAs (sRNAs) as key drivers of *Salmonella* stress survival. Generally thought to be 50–500 nucleotides in length and to occur in intergenic regions, sRNAs typically regulate protein expression through base pairing with mRNA targets. In this work, through employing a refined definition of sRNAs allowing for shorter sequences and sRNA loci to overlap with annotated protein-coding gene loci, we have identified 475 previously unannotated sRNAs that are significantly differentially expressed during carbon starvation (C-starvation). Northern blotting and quantitative RT-PCRs confirm the expressions and identities of several of these novel sRNAs, and our computational analyses find the majority to be highly conserved and structurally related to known sRNAs. Importantly, we show that deletion of one of the sRNAs dynamically expressed during C-starvation, sRNA4130247, significantly impairs the *Salmonella* C-starvation response (CSR), confirming its involvement in the *Salmonella* CSR. In conclusion, the work presented here provides the first-ever characterization of intragenic sRNAs in *Salmonella*, experimentally confirms that sRNAs dynamically expressed during the CSR are directly involved in stress survival, and more than doubles the *Salmonella enterica* sRNAs described to date.

## 1. Introduction

Over the past decade, the discovery of non-coding RNAs (ncRNAs) and the definition of their roles has been a prominent subject in the field of genetics. NcRNAs are found in all organisms and are known to regulate vital cellular functions and frequently act as master gene regulators [1,2]. An array of environmental stressors regularly affects cells and organisms throughout their life cycle, and the cellular response to such stressors require rapid, orchestrated modulations of gene expressions. Despite the majority of cellular stress survival research focusing on protein involvements to date, ncRNAs are increasingly being found to be instrumental driving forces behind stress-response mechanisms [1,2]. A diverse set of long and short ncRNAs (e.g., circaRNAs [3], miRNAs [4,5], and lncRNAs [6,7] in eukaryotes and 6S [8], CRISPR [9], and sRNAs [10] in prokaryotes) are now well characterized as serving specific roles during organismal and cellular stress protections and/or during persistent adaption.

The ability of certain species to withstand starvation for extended periods of time is a striking example of the capacity to adapt to and survive extreme environmental fluctuations. Bacterial small RNAs (sRNAs) are commonly associated with stress [9,10] and were recently hypothesized to coordinate *Salmonella* survival during carbon starvation (C-starvation) [11]. When *Salmonella* experience starvation of vital nutrients (e.g., carbon), the bacteria undergo several self-regulated genetic and physiological changes known as the starvation-stress response (SSR) to survive [12]. These metabolic changes include transformation and regulation of the creation of lipids, RNAs, proteins/enzymes [10]. That said, many of the genetic regulations and specific mechanisms driving the ability of *Salmonella* to survive C-starvation, remain undescribed.

Bacterial sRNAs are normally 50 to 500 base pairs long, frequently induced by stress, and primarily thought to function by binding to complementary mRNA target sequences to either enhance or inhibit their translation [13,14]. Over the last 10 years, next generation sequencing (NGS) has revolutionized the analysis of bacterial transcription. In fact, within just the last few years, several NGS studies have identified over 350 new sRNAs dynamically expressed in *Salmonella enterica* in response to stress (Appendix A) [11,15,16,17,18,19].

Notably, in 2016, our lab performed an RNA-Seq analysis of the *Salmonella enterica* serovar Typhimurium SL1344 transcriptomic response to C-starvation and successfully characterized 58 previously undescribed sRNAs dynamically expressed during the C-starvation response (CSR) [11]. That said, in this study, we reanalyzed these data using a combination of novel methodologies and refined criteria for sRNA annotation accounting for the identification of sRNAs less than 50 base pairs in length as well as sRNAs embedded within protein coding genes. Excitingly, this more thorough, well-contemplated approach culminated in the characterization of 475 additional, differentially expressed, previously uncharacterized small RNAs expressed from the *S. enterica* SL1344 genome. Importantly, in addition to their practical implications in terms of restricting outbreaks and potential to serve as therapeutic targets, our results strongly suggest that sRNA genes actually outnumber protein coding genes in *Salmonella* and potentially many other bacterial species.

## 2. Results

### 2.1. Identification of 475 Novel Salmonella sRNAs Dynamically Expressed during C-Starvation

To determine whether *S.* Typhimurium differentially expresses novel sRNAs in response to C-starvation conditions, (non-starved) log-phase, 5 h C-starved, and 24 h C-starved cell populations were generated. RNA was isolated from each culture and then sequenced using a small RNA RNA-seq protocol resulting in roughly 6 million paired-end 100 base pair (bp) reads for each culture. RNA-seq reads were uniquely aligned to the *S.* Typhimurium SL1344 genome, but in contrast to our previous study which required alignments to be >50 bp in length and to occur strictly within intergenic regions [11], we now allowed alignments to be ≥9 bp in length and to overlap (either partially or entirely) with annotated protein coding gene loci (Figure 1, Appendix A).

In addition, in order to identify sRNAs differentially expressed between the three growth conditions, the total number of reads aligning to each putative sRNA was calculated for each condition and regions exhibiting dynamic changes in expression selected. That said, whereas our previous study required at least a three-fold change in expression [11], in this work we required all sRNAs to exhibit a considerably less pronounced (≥10%) change in expression in response to C-starvation. Next, after identifying all putative sRNAs matching our refined criteria, we filtered all resulting putative sRNA sequences against the 396 previously reported *S.* Typhimurium SL1344 sRNAs [11,15,16,17,18,19] (Appendix A), and after removing 246 annotated sRNAs we find differentially expressed in response to carbon-source starvation, successfully characterize 475 novel, unannotated *Salmonella* sRNAs (Appendix A).

Importantly, small transcript Northern blotting of one the putative sRNAs identified in this work, sRNA4372553, agrees with our RNA-seq based expression analyses and confirms sRNAs less than 50 nt in length are indeed expressed in *S.* Typhimurium (Figure 2).

### 2.2. Characterization of sRNAs as Inter- or Intragenic

As sRNA loci are traditionally thought to occur within noncoding regions of the genome, we began by identifying all chromosomal intergenic regions longer than 29 bps from the SL1344 chromosome employing a methodology similar to our previous reports [11,15]. Out of the 475 novel sRNAs identified in this study, 368 are expressed from and wholly reside within intergenic sequences located between annotated gene loci (Appendix A). That said, in agreement with a recent study in *E. coli* identifying sRNA genes embedded within annotated protein coding loci [20], we also identified 107 differentially expressed sRNAs fully located within or partially overlapping 5′ or 3′ ends of a neighboring protein coding gene (Appendix A). Importantly, in order to distinguish the expression of sRNA sequences from overlapping genes, expressions of size-matched sequences corresponding to at least two distinct regions of the overlapping gene were determined (Figure 3).

### 2.3. sRNAs and Related Loci Are Highly Conserved

To examine the conservation of our novel sRNAs, we aligned our putative sRNAs to related bacterial genomes and find 312 are well conserved (Figure 4).

In addition to this, we also performed an in-depth examination of the relatedness of one of our highly expressed intragenic sRNAs (sRNA748945) to a sRNA recently reported as being embedded in a protein coding locus in *E. coli*. (*E. coli* sRNA703692) [20]. As both sRNAs sequences overlap the 5’ end of a neighboring protein coding gene (Figure 5A,B), we aligned the sequences of both the sRNAs and the protein coding genes (Figure 5C,D, respectively) as well as the predicted sRNA secondary structures [21] (Figure 5E), and found each to be strikingly conserved confirming sRNAs embedded in protein coding genes are indeed expressed in S. Typhimurium.

### 2.4. sRNA Profiles Associated with Distinct States of C-Starvation Are Highly Distinct

Of the 475 novel sRNAs, unsupervised hierarchical clustering [22] of the sRNA expressions identified three principle clusters of expression. Nearly two out of three of all of the novel sRNAs were most highly expressed during log phase (non-starved) making up the largest cluster of sRNAs. For the intragenic sRNAs, 66 sRNAs were most highly expressed during log phase, while 26 and 15 sRNAs were predominately expressed during 5 and 24 h C-starvation, respectively (Figure 6A). For the intergenic sRNAs, 231 sRNAs made up the log phase cluster, whereas 106 and 31 sRNAs were most highly expressed during 5 and 24 h C-starvation, respectively (Figure 6B).

Importantly, we find unsupervised hierarchical clustering of the expressions of the 396 previously annotated sRNAs similarly identifies three principal clusters of sRNA expressions each being predominately comprised of sRNAs almost exclusively expressed at one of the three distinct states of C-starvation examined (Appendix A).

### 2.5. Identification of Putative sRNA Targets

As sRNAs are typically thought to function via antisense binding to target mRNAs [23], we next asked whether the sRNAs identified in this study harbored any significant sequence complementarities to known SL1344 mRNAs. Strongly supporting the functional relevance of the our sRNAs, we find that ~54% of our putative sRNAs either overlap the 5’ or 3’ ends of neighboring coding DNA sequences (CDS): mRNA, rRNA or tRNA loci (Appendix A); or share significant sequence complementarities to known mRNAs transcribed elsewhere in the SL1344 genome (Appendix A) [24] suggesting the principle function of the majority of the sRNAs identified in this work is to regulate transcripts via antisense base-pairing (Appendix A).

### 2.6. Deletion of Novel sRNAs Can Significantly Impair the Salmonella Starvation Response

To further explore potential functional roles for sRNAs dynamically expressed during the CSR, we selected sRNA4130247 and sRNA4720054 for deletion analysis, as we found both differentially and significantly expressed during the CSR and could successfully confirm their dynamic expressions via qRT-PCR (Appendix A).

Lambda-Red recombinase-based methodology [25] was used to delete the loci corresponding to each of these sRNAs (as well as that of the previously annotated STnc1200 sRNA locus [18]) resulting in the creation of two mutant Salmonella strains: ∆413 and ∆472 (and a STnc1200 mutant we termed ∆92 as a control) for subsequent phenotypic testing. Deletions were verified by PCR confirmation of sRNA (~100–200 bp) replacement with a chloramphenicol resistance cassette (1034 bp) (Figure 7A).

As these sRNAs were initially identified due to their dynamic expression in response to C-starvation, we began by assessing the growth of sRNA deletion mutants in normal and low Carbon media. Importantly, whereas we found the growth of each our mutant strains largely indistinguishable from that of WT Salmonella under normal conditions, and that ∆92 and ∆472 growth curves remained similarly unaffected by growth in low Carbon media, we found ∆413 growth to be significantly impaired nearing total Carbon depletion after 8 h in low Carbon media (*p* = 0.0023) (Figure 7B). In light of this, we next asked whether sRNA4130247 and/or sRNA4720054 might similarly be involved with survival during other forms of stress electing to examine the effects of our deletions on heat (55 °C) and Polymyxin B exposure survival. Interestingly, we found both heat and Polymyxin B survivability to be significantly enhanced for both ∆413 and ∆472 sRNA deletion mutants following C-starvation as compared to controls (WT and ∆92). Notably, while we found deletion of either sRNA actually enhanced survivability during heat challenge, we found ∆472 survival to be the most pronounced averaging 69.1% cell survival following heat challenge versus 7.0% WT cell survival (*p* = 0.0003) (Figure 7C), and importantly, that sRNA expression vector transformation can restores WT-like survivability (Appendix A). Similarly, we also found deletion of either sRNA enhanced survivability during Polymyxin B exposure, but that ∆413, not ∆472, was the most significantly enhanced surviving ~2.5 times better than WT (*p* = 0.0147) (Figure 7D).

In addition to these effects, we also elected to determine whether any sRNAs dynamically expressed during the CSR might participate in biofilm formation. In all we selected eight sRNAs and began by determining the effects of sRNA deletion on Curli production as Curli is an essential biofilm component [26]. While we found neither the deletion nor overexpression of sRNA4130247, sRNA4720054, or five other sRNAs significantly affected Salmonella biofilm formation, we did find deletion of sRNA1186573 all but eliminated Curli production and decreased biofilm production to ~4% of WT (Appendix A), and that this effect was significantly rescued upon sRNA1186573 overexpression (Appendix A).

## 3. Discussion

Our group previously characterized 58 sRNAs dynamically expressed in response to C-starvation [11], and now using refined criteria, have identified an additional 475 novel small RNAs likewise dynamically expressed in response to Carbon starvation. Three hundred and sixty-eight of these to intergenic small RNAs, and 107 to intragenic sRNAs. To ensure intragenic sRNAs represented distinct transcripts, we determined the expression of multiple regions of the gene overlapping with a given sRNA and confirmed that the expressions significantly differed (Figure 3). Excitingly, this work represents the first characterization of intragenic sRNAs in *Salmonella*. Importantly, our intragenic sRNA work strongly agrees with the 1st reported intragenic sRNAs discovered in *E*. *coli* in 2018 [20]. Of note, we find several of the 58 intragenic sRNAs reported by Dar et al. to be well conserved in *Salmonella* (Figure 4) and have now similarly identified 107 intragenic sRNAs dynamically expressed in response to C-starvation in *Salmonella* suggesting that intragenic sRNAs are likely more prevalent than currently appreciated. That said, we have purposefully refrained from classifying sRNAs into existing annotations (e.g., asRNA, cis-RNAs, decRNAs, trans-RNAs) [20] as we feel thorough evaluation of each individual sRNA is beyond the scope of this initial manuscript, and defining the mechanisms driving individual sRNA expressions and their functional roles are better left to more focused evaluations.

Furthermore, we find several lines of reasoning suggesting that *Salmonella* sRNA loci likely actually outnumber that of protein coding genes. (1) Strikingly, we find 2620 of 3035 intergenic SL1344 chromosomal regions exclusively align to at least 10 unique small RNA reads generated in our initial small RNA-seq analyses. (2) The realization and experimental verification that numerous additional sRNA loci are embedded in that of protein coding loci. (3) The realization and experimental verification that sRNAs less than 50 nt in length are indeed expressed in *S.* Typhimurium. In all, this report describes 99 novel sRNAs with proposed lengths between 29 and 49 nts. As several previous studies aimed at novel *Salmonella* sRNA characterization required minimal sRNA lengths of 50 nt or more for inclusion [11,27], we suggest many sRNAs <50 nt in length have likely simply been overlooked to date. Additionally, (4) while the sRNAs expressed in response to acid [20], starvation [11], and during virulence [18] (Appendix A) are highly distinct, our findings suggest that sRNA profiles differ dramatically during different stages of individual environmental challenges. Unsupervised hierarchical clustering of the expressions of the 475 novel sRNAs discovered in this work (as well as the 396 previously annotated *Salmonella* sRNAs) identifies three principal clusters of genes—defined as being predominately comprised of sRNAs almost exclusively expressed at one of the three distinct states of C-starvation examined. Of the 475 novel sRNAs, 297 genes constitute the largest cluster being characterized by expression largely restricted to log (non-starved) phase. Similarly, 132 sRNAs are primarily expressed at 5 h of carbon starvation, and 46 at 24 h of carbon starvation (Figure 6). This demonstrates that not only are the sRNAs expressed during log phase growth and short-term starvation highly distinct, but much to our surprise, the sRNAs expressed during initial/short-term starvation are similarly also highly distinct from those expressed during more long-term/prolonged starvation.

Importantly, while we find the growth of several sRNA deletion strains largely indistinguishable from that of WT *Salmonella* under normal conditions, and that the growth curves of the majority of these strains remain similarly unaffected in low-carbon media, we do find growth of a sRNA4130247 mutant significantly impaired after 8 h in low Carbon media (Figure 7A) supporting an active role for this sRNA and likely others identified in this work in the CSR. That said, whereas we find ∆413 growth significantly impaired in low-carbon media, we find both heat and polymyxin B survivability is significantly enhanced for ∆413 and ∆472 sRNA mutants (as compared to WT). Intriguingly, however, we find ∆413 and ∆472 mutants only exhibit enhanced heat and polymyxin B survivability after being subjected to C-starvation (Figure 7B,C). Notably, we also find ∆413 and ∆472 sRNA colonies frequently vary in size and that 10 to 20% of these are markedly smaller than most others from the same sample (data not shown). As small colony variants (SCVs) and enhanced survivability are both potential indicators of increased persister cell formation [28], we suggest sRNA4130247 and/or sRNA4720054 may be involved in regulating the decision between active stress resistance and persister formation. Furthermore, we find the sRNA transcriptome profiles of 5 and 24 h starved cells to be highly distinct with few commonalities (Figure 6) and similarly observe highly distinct sRNA profiles associated with short duration (24 h/recently fully desiccated) and prolonged (72 h) desiccated cells [15]. Interestingly, however, we find significant overlaps between the sRNAs expressed during short duration desiccation and those expressed after 5 h of C-starvation, and similarly find significant overlaps between the sRNAs expressed during prolonged desiccation and those expressed after longer 24 h C-starvation [11,15]. That said, as sRNAs are clearly vital to efficient cellular stress survival, sRNA participation in persister cell formation will undoubtedly be explored in the near future.

In conclusion, the 475 novel sRNAs characterized in this report more than double the number of *Salmonella enterica* serovar Typhimurium sRNAs described to date, increasing the total number to characterized *Salmonella* sRNAs from 396 to 871 (Appendix A). In addition to this, this work also describes the first-ever characterization of intragenic sRNAs in *Salmonella* and experimentally confirms that deletion of one of the sRNAs dynamically expressed during C-starvation, sRNA4130247, significantly impairs the *Salmonella* C-starvation response, confirming its involvement (and suggesting the involvements of many other sRNAs identified in this work) in starvation survival. That being said, we also find significant evidence suggesting there are many additional *Salmonella* sRNAs yet to be discovered and that sRNA loci are comparable to and may actually outnumber protein-coding genes. Importantly, although the experimental characterization of endogenous roles for *Salmonella* sRNAs will take time, their potential utility as novel pharmacological targets and/or agents for use in combatting antibiotic resistant strains and eliminating persistent *Salmonella* infections is extremely promising.

## 4. Materials and Methods

### 4.1. Bacteria Strains and Media

The bacterial strain examined in this project was *Salmonella enterica serovar* Typhimurium strain SL1344 (*hisG46*). Cultures were grown in 22.5 mM potassium phosphate and 10 mM ammonium chloride MOPS-buffered salts (MS) medium with 0.2 mM histidine. Log-phase cells were grown in an MS medium that contained 0.4% (w v-1)glucose (MS hiC). C-starved cells were cultivated in MS medium with 0.03% (w v-1) glucose (MS loC).

### 4.2. Cultivation of Non-Starved Log-Phase and C-Starved Cell Cultures

SL1344 was grown overnight in MS hiC medium at 37 °C with constant shaking (18–24 h). Overnight cultures were incubated at 37 °C with aeration/shaking after being diluted 1:100 with fresh MS loC and fresh MS hiC medium. Culture growth was monitored at 600 nm using a standard spectrophotometer. MS hiC cultures were triggered at an OD600 of 0.3–0.4 to produce log-phase cells. MS hiC cultures continued to grow until OD600 ceased increasing due to depletion of glucose. Cultures were then C-starved for 5 and 24 h to produce 5 h C-starved and 24 h C-starved *Salmonella* cells, respectively [11].

### 4.3. RNA Isolation and Small RNA Sequencing

RNA was isolated from each of the SL1344 cultures (log-phase, 5 and 24 h C-starved) using Trizol^®^ (Thermo Fisher Scientific, Waltham, MA, USA) per standard manufacture protocol then shipped to Otogenetics (Atlanta, GA, USA) for commercial RNA-seq using an Illumina MiSeq genome sequencer and small RNA protocol (paired ends, 100 bp, 25M reads). Otogenetics employed the NEBNext Multiplex Small RNA Library Prep Set for Illumina (NEB, Ipswich, MA, USA) coupled with automated agarose gel size selection (30 to 200 nt) using the Pipin Prep Instrument (Sage Science, Beverly, MA, USA) for small RNA library prep. Adapter sequences were removed with Cutadapt [29].

### 4.4. Identification of sRNA Consensus Sequences

The current Ensembl SL1344 genome [30] was aligned to individual RNA-seq reads using BLAST+ (2.2.27) [31]. A “best hit” parameter was employed to ensure that individual reads aligned to a single genomic location. All resulting alignments were filtered by requiring a minimum length of 29 base pairs and 85% sequence identity. The number of reads aligning to each unique region was determined in each of the three growth conditions (log, 5 and 24 h C-starved), and putative sRNAs further defined as regions aligning to at least 200 unique RNA-seq reads and demonstrating at least a 10% change in expression between growth conditions. sRNA sequences were refined using the graphical viewer Tablet, where the indexed BAM files containing RNA-seq reads (found using the annotated SL1344 reference genome) allowed the visualization of regions of interest and evaluation of identical or overlapping reads [32]. Chromosomal position of peak expression within each resultant region of interest was determined and chromosomal positions (upstream and downstream) where peaks fell below 75% of the maximum were identified in each sample. Start and stop positions in each of the three samples where a putative sRNA was expressed were averaged resulting in the final sRNA consensus sequence calls. Putative sRNA genomic loci defined as overlapping either in part or in entirety with any existing Ensembl annotated SL1344 gene [33] were additionally required to differ in expression from overlapping genes by ≥200%. When sRNAs were defined as partially overlapping one terminus of an annotated locus, two 50 bp regions were extracted from the overlapping gene (one from the gene center and one from the opposing terminus). When sRNAs were defined as occurring entirely internal to an annotated locus, two non-overlapping unique 50 bp regions were extracted from the overlapping gene at random positions. Next, extracted overlapping gene 50 bp sequences were aligned to RNA-seq reads and expressions determined identically as calculated for putative sRNA loci.

### 4.5. sRNA Computational Analyses and Target Gene Prediction

NCBI BLAST megablast tool (word size: 16) was used to search for homologous sequences in all available bacterial genomes (excluding *Salmonella*) [30]. SRNAs were considered conserved if genomic sequences with ≥80% identity to the full length of a novel sRNA were identified in at least one other bacterial genus. Mfold was used to predict sRNA probable secondary structures [21]. Potential targets for each novel sRNA (Appendix A) were identified by both IntaRNA (rna.informatik.uni-freiburg.de) and alignment to current SL1344 genes annotated by Ensembl using BLAST+ (2.2.27) as previously reported [31].

### 4.6. Small Transcript Northern Blotting

Total RNA from SL1344 cultures was isolated with Trizol^®^ (Thermo Fisher Scientific) per standard manufacture protocol. A 15% acrylamide/bis-acrylamide (29:1) gel containing 8 M urea (48% (*w*/*v*)) and 1× TBE was prerun for 30 min at 100 V in a vertical mini- PROTEAN tank (Bio-Rad). Gels were flushed and loaded with 10 µg of total RNA in 2× TBE/Urea sample buffer (Bio-Rad), then run at 200 V until the bromophenol blue dye front reached the gel bottom. As a size reference, 1 µL of pooled, commercially synthesized biotin 5’ end–labeled DNA oligonucleotides (25, 50, and 200 bp each at 1 µM) was also loaded in 2× TBE/Urea sample buffer. After removal from the electrophoresis plates, gels were gently rinsed with water then washed in 0.5× TBE for 5 min on an orbital shaker. After electrophoresis, RNA was electro-transferred (Mini Trans-Blot Electrophoretic Transfer Cell apparatus, Bio-Rad) to Biodyne B Pre-cut Modified Nylon Membranes 0.45 µm (Thermo Fisher Scientific) for 2 h at 20 V in 0.5× TBE. After removal from the transfer stack, membranes were gently washed in 1× TBE for 15 min on an orbital shaker, then UV cross-linked at 1200 mJ for 2 min (Stratalinker, Stratagene). Prehybridization was performed in North2South^®^ Hybridization Buffer (Thermo Scientific) at 42 °C for 30 min, after which 30 ng (per milliliter of hybridization buffer) of each appropriate biotin 5’ end–labeled oligonucleotide was added directly to the hybridization buffer as probe.Probe4372453         5pBio-TGCGACGTTAAGAATCCGTATCTProbe4372453RevComp   5pBio-AGATACGGATTCTTAACGTCGCAProbe4372453SCRAMBLE  5pBio-CGTAGTGAATACCCGTTTATACG

### 4.7. Deletions via Lambda-Red Protocol

To analyze the effects of the deletion of the sRNAs in the genome of *Salmonella*, we used the Lambda-Red recombines method with some modifications, as following [15]. Strain EC118 (*E. coli* + pKD3–Cm^r^) was grown overnight at 37 °C in the presence of chloramphenicol with shaking. Plasmids were isolated using the Wizard Plus mini-prep kit and used as a template for PCR reaction to obtain the chloramphenicol (Cm) resistance cassette. A PCR was performed with a pair of primers containing 20 nt sequence for amplification of the Cm resistance gene plus a 40 nt sequence homologous to the 5’ or 3’ ends of the target gene. After PCR verification in agarose gel and PCR clean-up using the ISOLATE II PCR and Gel Kit (Bioline), the remaining plasmid template was digested with *Dpn*I at 37 °C for 3 h. The final PCR product was used for electroporation in competent SL1344 cells (carrying pKD46 plasmid) in a 0.1 cm cuvette at 1.8 kV setting. After electroporation, the cells were inoculated in SOC-pyruvate broth (2% tryptone, 0.5% yeast extract, 10 mM MgCl_2_, 10 mM MgSO_4_, 20 mM glucose, 0.3% pyruvate) and incubated at 37 °C with shaking for 3 h. The recombinants were concentrated by centrifugation and grown on Luria-Bertani (LB) agar (Tryptone 1%, NaCl 1%, Yeast extract 0.5%, Agar 1.5%, pH 7.0) containing 45 µg/mL of Cm. The colonies were tested for replacement of target gene by Cm resistance cassette by PCR using the following primers: 413f (AGCCAAGATGCAAGAATAGACA)/413r (CCACGCTAATCACGACCA) or 472f (TTACTTACCGGAGGCGACAT)/472r (GAAAATTCTCCATCGCGG). Strain SMS957 (carries pCP20 plasmid) was inoculated into 2 mL LB-Amp broth and incubated at 30 °C for 24 h with shaking. pCP20 (Flp recombinase gene under control of temperature-sensitive (Ts) promoter; Ts replicon) [34] was isolated using plasmid mini-prep kit and used for electroporation into competent cells, prepared as in 3.4.2. After transformation, the cells were added to SOC media (containing 1 mM sodium pyruvate) and incubated at 30 °C for 2 h with slow shaking. The cells were concentrated by centrifugation and plated on LB-Amp agar and incubated at 30 °C overnight. Six colonies were streaked into LB agar without antibiotics and incubated at 43 °C overnight. After growing, the strains were streaked on LB + Amp and LB + Cm to confirm the excision of both antibiotic resistance gene from the genome and excision of the plasmids. The strains that were negative for both Cm and Amp (no growth on any antibiotics) were selected and stored at −80 °C for further analysis.

### 4.8. Quantitative RT-PCR

Total RNA was isolated from control (24 hr water) and desiccated samples (24 h and 72 h) using the RNeasy PowerMicrobiome Kit (Qiagen, Germantown, MD, USA). qPCR was performed using the iTaq^TM^ Universal SYBR^®^ Green One-Step Kit (Bio-Rad, Hercules, CA, USA) according to the manufacturer’s instructions. Five annotated genes were selected to validate the small RNA-Seq data: *STnc3920*, *STnc1460*, *STnc700*, *IsrL* and *sRNA294324*. Five previously unannotated genes were also selected for validation: sRNA3981754, sRNA3417670, sRNA1320429, sRNA3417448, and sRNA294677. Primers are listed in Appendix A. The gene *rpoD* was used as a control to normalize the values as it has shown not to be differentially expressed under these conditions. The reactions were performed in duplicate in a 384-well plate containing 5 μL of iTaq universal SYBR^®^ Green reaction mix, 0.125 μL of the iScript reverse transcriptase, 300 nM of each forward and reverse primers, 45 ng of RNA and nuclease-free water to a total reaction mix volume of 10 μL. qRT-PCR was conducted in the CFX384 Touch Real-Time PCR Detection System (Bio-Rad). The real-time PCR program was as follows: initial reverse transcription reaction at 50 °C for 10 min, polymerase activation and DNA denaturation at 95 °C for 1 min, followed by 40 cycles of amplification at 95 °C for 10 s and 60 °C for 15 s. All the PCR amplifications were performed in triplicate. DNase-treated RNA was used in all sample extractions, and a control without reverse transcriptase was included. Specificity of amplifications was verified using melting curves. Gene expression was calculated via the Delta-Delta cycle threshold method [35].

### 4.9. Phenotypic Stress Challenges

#### 4.9.1. Biofilm Formation and Curli Fimbriae Production

As some of the selected sRNAs were predicted to be related to biofilms, the strains were subjected to biofilm formation assay in 96-well polystyrene plates. Cultures were grown overnight in tryptic soy broth (TSB), and 20 µL from the overnight cultures were added to 130 uL 1× TSB (non-starved state) or 20× diluted TSB (1/20 TSB, for starved state) [36] for 15 h (non-starved state), and up to 48 h (starved state). Tests were performed in quadruplicate. After inoculation, the plates were sealed and incubated at room temperature (RT, 23 °C) without shaking. A sterile needle was used to make a small hole in the center of each well to allow oxygen exchange and was compared to the completely sealed plate, which was in microaerophilic environment. Following RT incubation, the inoculum was discarded by inverting the plates and the wells were washed 3× with distilled water. Then, 170 µL of 75% ethanol was added to each well and incubated at RT for 10 min. The ethanol was discarded, and the plate was dried for 20 min at 37 °C. Crystal violet (0.1% *w*/*v*) was added to the wells (170 µL/well) and incubated at RT for 10 min. After incubation, the crystal violet was discarded, and the plates were washed in running water until no excess of the dye was seen in the plate. Finally, 170 µL of glacial acetic acid were added to each well and the plate was read in a Synergy microplate reader (BioTek, Winooski, VT, USA) at Abs = 570 nm. For curli fimbriae production, we followed the method used in a previous work on bacterial sRNA related to biofilm formation (Bak et al., 2015). The strains were streaked onto LB/Congo Red plates (LB agar without NaCl, 40 µg/mL Congo red and 20 µg/mL Coomassie Blue) and incubated at RT for 48 h. Colonies displaying pink-white color were considered curli fimbriae negative.

#### 4.9.2. Growth Curve Assay

To assess the effects of manipulating levels of selected sRNAs on growth rate, overnight cultures of mutant and wildtype strains were inoculated (1:100) into MS-hiC and MS-loC media. All samples were grown at RT with intermittent shaking and absorbance (OD = 600 nm) measurements were recorded every 30 min for 16 h. All measurements were taken on the same Synergy microplate reader as above.

#### 4.9.3. Peroxide/Poly B Assays

To assess tolerance to heat and hydrogen peroxide challenges, deletion mutants and wild type *Salmonella* were exposed to these conditions under the following parameters. Cultures were grown overnight in LB broth with shaking at 37 °C. The next morning, non-starved overnight cultures were inoculated 1:100 into fresh hiC-MS media and grown to an OD_600_ of approximately 0.3–0.4 to generate log-phase cells. Starved cultures were inoculated in same ratio into loC media, allowed to reach stationary phase and subsequently kept on the shaker for additional 3 h to become starved. Aliquots of all cultures were serially diluted (1:5) in sterile distilled water and plated on LB agar to serve as control. For challenge with hydrogen peroxide, 10 µL of H_2_O_2_ (1M) and 10 µL of log-phase culture were added to 980 µL of fresh LB broth and incubated at 37 °C with shaking for 45 min. For heat tolerance assessment, cultures were incubated at 55 °C with shaking for 30 min. Following each treatment, aliquots from each sample were serially diluted and plated in the same manner as control, and all plated samples were incubated at 37 °C overnight. CFUs were then determined and used to calculate survival under each condition.

## Figures and Tables

**Figure 1 antibiotics-10-00305-f001:**
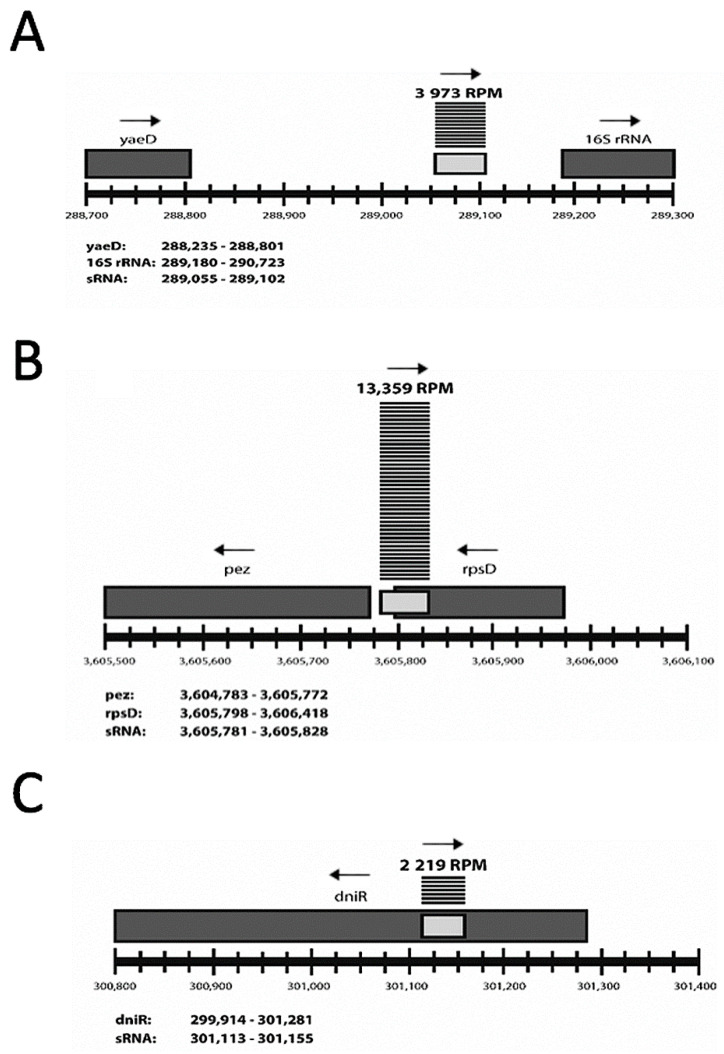
Three classes of small RNAs as defined by relative position to nearest annotated neighbor. Examples of (**A**) intergenic, (**B**) overlapping, and (**C**) internal sRNA loci. Light gray boxes represent putative sRNAs. Dark gray boxes represent neighboring annotated genes. Horizontal lines above sRNAs represent unique sequence reads mapping to that locus during log phase with the total reads per million (RPM) indicated above. Chromosomal start and stop positions for each neighboring gene and sRNA are listed. Transcriptional direction is indicated by an arrow just above a sRNA/gene name.

**Figure 2 antibiotics-10-00305-f002:**
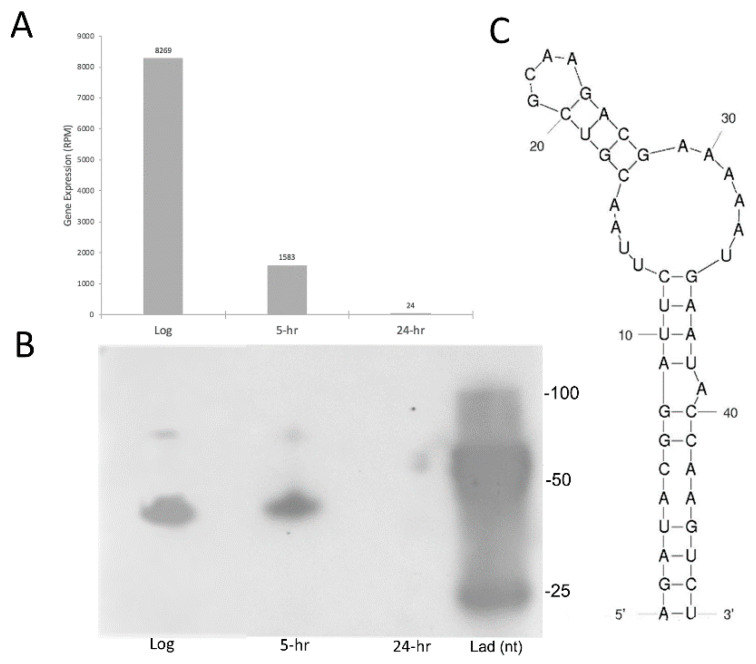
Northern blot validation of sRNA differential expressions. (**A**) The gene expression of sRNA4372453 depicts a decrease in expression from non-starved log phase to 24 h C-starvation. (**B**) Small transcript northern blot of sRNA4372553. Lad (nt), oligonucleotide ladder with band sizes indicated to the right. Log, log phase. 5-h, 5 h C-starved. 24-h, 24 h C-starved. (**C**) Predicted secondary structure [21] of sRNA4372453.

**Figure 3 antibiotics-10-00305-f003:**
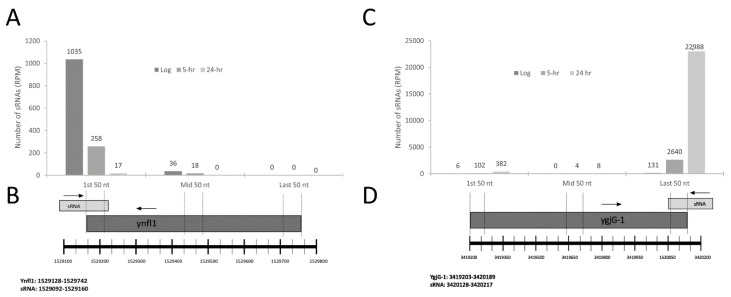
Overlapping sRNAs and protein coding genes have distinct expressions. (**A**) Gene expressions measured at distinct locations within the shared sRNA1529092/ynfl1 locus. (**B**) Illustration depicting locations aligned to RNA seq data to determine expressions shown in (**A**). (**C**) Gene expressions measured at distinct locations within the shared sRNA3420128/ygjG-1 locus. (**D**) Illustration depicting locations aligned to RNA seq data to determine expressions shown in (**C**). Chromosomal start and stop positions for each protein coding gene and sRNA are listed. Transcriptional direction is indicated by an arrow just above a sRNA/gene name.

**Figure 4 antibiotics-10-00305-f004:**
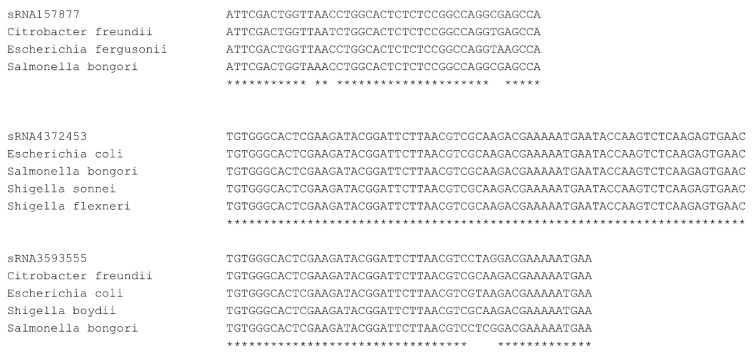
ClustalW alignments of novel sRNAs with conserved loci. (**Top**) sRNA157877 (**Middle**) sRNA4372453 (**Bottom**) sRNA3593555. *, 100% nucleotide identity.

**Figure 5 antibiotics-10-00305-f005:**
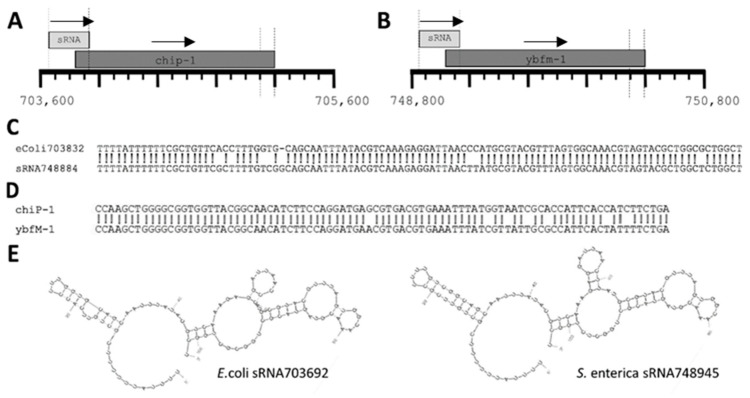
*E. coli* sRNA703832 and Salmonella sRNA748884 sequence comparison. (**A**) Cartoon illustrating genomic position of sRNA703692 and overlapping protein coding gene in Escherichia coli. (**B**) Cartoon illustrating genomic position of sRNA748945 and overlapping protein coding gene in Salmonella enterica SL1344. (**C**) ClustalW alignment of *E. coli* sRNA703692 and *S. enterica* sRNA748945 genomic loci as demarcated by vertical lines in A and B. (**D**) ClustalW alignment of 3’ ends of *E. coli* chip-1 and *S. enterica* ybfm-1 genomic loci as demarcated by vertical lines in A and B. (**E**) Predicted secondary structures [21] of *E.coli* sRNA703692 (**left**) and *S. enterica* sRNA748945 (**right**). Transcriptional direction is indicated by an arrow just above a sRNA/gene name.

**Figure 6 antibiotics-10-00305-f006:**
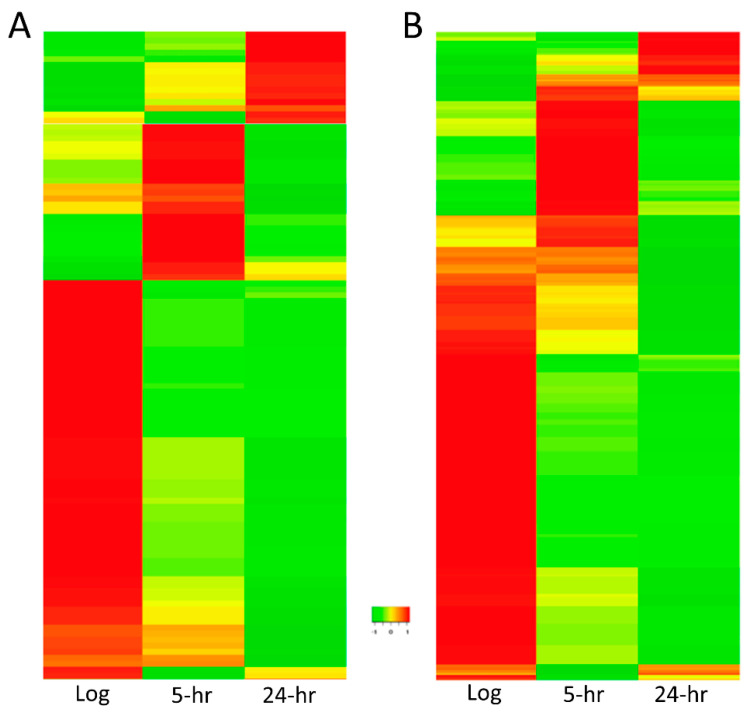
Heat maps depicting gene expressions of novel sRNAs. (**A**) Heatmap [22] of intragenic sRNAs. (**B**) Heatmap of intergenic sRNAs. Log, log phase; 5-h, 5 h C-starved; 24-h, 24 h C-starved.

**Figure 7 antibiotics-10-00305-f007:**
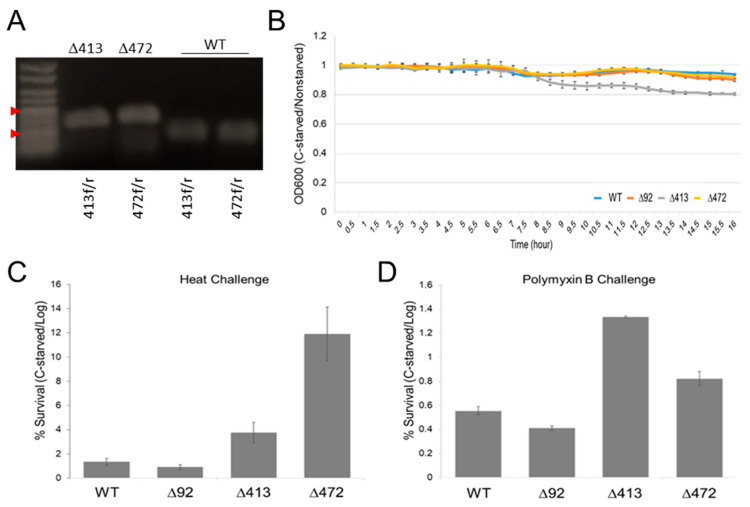
Effects of silencing sRNA4130247 and sRNA4720054. (**A**) PCR confirmation of sRNA genomic replacement with chloramphenicol resistance cassette. Arrowheads indicate 1000 and 200 bp ladder bands. Lane 1, lkb ladder. Lanes 2–3, mutant colonies (harboring ~1000 bp insertions at indicated loci). Lanes 4–5, wild type sRNA amplicons (<200 bp). Primer sets are indicated below image. (**B**) Growth of deletion mutants and wild type Salmonella in normal and low-carbon media. OD600 ratio (C-starved/nonstarved) determined every 30 min for 16 h. (**C**) Ratio of C-starved/nonstarved cell survival of log phase deletion mutants and wild type Salmonella subjected to 55 °C for 30 min. (**D**) Ratio of C-starved/nonstarved cell survival of deletion mutants and wild type Salmonella during Polymyxin B exposure. Error bars in B-D indicate SD (n = 3). Survivals were normalized to wild type (WT). ∆413, sRNA4130247 deletion mutant; ∆472, sRNA4720054 deletion mutant; ∆92, STnc1200 sRNA deletion mutant.

## Data Availability

The data presented in this study are openly available in the NCBI Sequence Read Archive (SRA) repository under SRA accession number SRP058591.

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
