# Peer review of "Characterization of 475 Novel, Putative Small RNAs (sRNAs) in Carbon-Starved Salmonella enterica Serovar Typhimurium"

_antibiotics, 2021, doi:10.3390/antibiotics10030305_

Round 1

Reviewer 1 Report

In this work the authors of the article proposed the first characterization of intragenic sRNAs in Salmonella and described 99 novel sRNAs with proposed lengths between 29 and 49 nucleotides. The phenotype after the deletion of two sRNAs independently was investigated. The results of the work may have practical application as a as novel pharmacological targets.

I have one comment to the article: please, add the results of the sRNA deletion confirmation in the supplemental data. 

Author Response

Reviewer 1 - Comments and Suggestions for Authors

In this work the authors of the article proposed the first characterization of intragenic sRNAs in Salmonella and described 99 novel sRNAs with proposed lengths between 29 and 49 nucleotides. The phenotype after the deletion of two sRNAs independently was investigated. The results of the work may have practical application as a as novel pharmacological targets.

We would first like to thank the reviewer for their thoughtful, largely positive evaluation of our work.

I have one comment to the article: please, add the results of the sRNA deletion confirmation in the supplemental data.

We agree with the reviewer this warrants inclusion and have therefore modified the text and materials/methods accordingly and Figure 7 as follows:

Figure 7. Effects of silencing sRNA4130247 and sRNA4720054. (a) PCR confirmation of sRNA genomic replacement with chloramphenicol resistance cassette. Arrowheads indicate 1,000 and 200 bp ladder bands. Lane 1, lkb ladder. Lanes 2-3, mutant colonies (harbouring ~1,000 bp insertions at indicated loci). Lanes 4-5, wild type sRNA amplicons (<200 bp). Primer sets are indicated below image. (b) Growth of deletion mutants and wild type Salmonella in normal and low-carbon media. OD600 ratio (C-starved/nonstarved) determined every 30 minutes for 16 hours. (c) Ratio of C-starved/nonstarved cell survival of log phase deletion mutants and wild type Salmonella subjected to 55°C for 30 minutes. (d) Ratio of C-starved/nonstarved cell survival of deletion mutants and wild type Salmonella during Polymyxin B exposure. Error bars in B-D indicate SD (n = 3). Survivals were normalized to wild type (WT). * indicates p ≤ 0.05; p-values determined by unpaired two-tailed t-test. ∆413, sRNA4130247 deletion mutant; ∆472, sRNA4720054 deletion mutant; ∆92, STnc1200 sRNA deletion mutant.

Reviewer 2 Report

In this manuscript, the authors analyze and classify the sRNAs of S. enterica with a length <50 nt and indagate the relationship between C-starvation and the expression of these sRNAs. The aim of the study, the experimental methods, and the conclusion are consistent. The experimental validation of the RNA-seq data with other experimental approaches (Northern blot and qRT-PCR) makes the manuscript complete and constitutes the real strength of the investigation strategy. However, the manuscript has some weaknesses related to the method of classification and definition of the sRNAs. In addition, some points should be clarified both to improve readability and to make the topic more attractive for readers. For this, I suggest that this manuscript can be accepted after major revisions.

  1. According to my opinion, the introduction can be expanded by adding information on other studies focused on ncRNAs of Gram-negative and Gram-positive bacteria, this paper will be more general and more attractive for microbiologists. Also, the authors could add information in the discussion, with a particular focus on the stress response and the ncRNAs expression. Finally, what about the relation between sRNA-targeted genes and virulence? By adding clinical details in discussion this paper will be more attractive for clinicians.
  2. It is not clear the method used for the classification of the sRNAs overlapped with coding sequences. Are these sRNAs oriented in an antisense manner with the genes? Or, as an alternative, are these RNAs produced by RNA decay (decRNAs) starting from a coding-protein mRNA? Please, clarify this point. I also suggest following another scheme of classification. For example, the ncRNAs that overlap with one gene and showing an antiparallel orientation are called asRNAs (antisense RNAs) or cis-RNAs, while the ncRNAs produced by RNA decay are called decRNAs. On the contrary, the ncRNAs coded from an intergenic sequence are called trans-RNA. I suggest renaming the classes of sRNA to improve the readability.
  3. In the genetic maps of figures, the strand of transcription of genes and sRNAs are not clear, so it is difficult to understand the correct orientation of the transcriptional unit. I suggest adding arrows representing the direction of the transcription and the tags 3’ or 5’ for the extremities (figures 1, 3, and 5).
  4. What is the name of 4th line of Figure 2 panel B? Is it the marker? Please indicate it (in figure or/and in the caption). I also suggest indicating the MW near the arrow and, with another arrow, the MW of the sRNA.
  5. The authors used IntaRNA and BLAST to predict the function of intergenic sRNAs, but it is not clear where the results are reported. Add these details in the caption of supplemental files.
  6. The names of the Bacteria should be written in italics and the names of the species can be with the first lowercase letter. Please check all the text for this issue.

Author Response

Reviewer 2 - Comments and Suggestions for Authors

In this manuscript, the authors analyze and classify the sRNAs of S. enterica with a length <50 nt and indagate the relationship between C-starvation and the expression of these sRNAs. The aim of the study, the experimental methods, and the conclusion are consistent. The experimental validation of the RNA-seq data with other experimental approaches (Northern blot and qRT-PCR) makes the manuscript complete and constitutes the real strength of the investigation strategy. However, the manuscript has some weaknesses related to the method of classification and definition of the sRNAs. In addition, some points should be clarified both to improve readability and to make the topic more attractive for readers. For this, I suggest that this manuscript can be accepted after major revisions.

We would first like to thank the reviewer for their thorough and largely positive evaluation of our work.

  1. According to my opinion, the introduction can be expanded by adding information on other studies focused on ncRNAs of Gram-negative and Gram-positive bacteria, this paper will be more general and more attractive for microbiologists. Also, the authors could add information in the discussion, with a particular focus on the stress response and the ncRNAs expression. Finally, what about the relation between sRNA-targeted genes and virulence? By adding clinical details in discussion this paper will be more attractive for clinicians.

We agree discussion of ncRNA expression during stress response and potential clinical implications warrants further discussion and could potentially broaden interest in our work, and have therefore added the following to the Discussion:

As small colony variants (SCVs) and enhanced survivability are both potential indicators of increased persister cell formation [29], we suggest sRNA4130247 and/or sRNA4720054 may be involved in regulating the decision between active stress resistance and persister formation. What’s more, we find the sRNA transcriptome profiles of 5 and 24 hour starved cells to be highly distinct with few commonalities (Figure 6) and similarly observe highly distinct sRNA profiles associated with short duration (24 hr / recently fully desiccated) and prolonged (72 hr) desiccated cells [15]. Interestingly, however, we find significant overlaps between the sRNAs expressed during short duration desiccation and those expressed after 5 hours of C-starvation, and similarly find significant overlaps between the sRNAs expressed during prolonged desiccation and those expressed after longer 24hr C-starvation [11,15].  That said, as sRNAs are clearly vital to efficient cellular stress survival, sRNA participation in persister cell formation will undoubtedly be explored in the near future.”

That said we do, however, respectfully disagree with the reviewer as to the appropriateness of “adding information on other studies focused on ncRNAs of Gram-negative and Gram-positive bacteria” to the introduction as we feel it too broad an area to suitably cover and to do so would distract the reader from the focus of our paper – novel sRNA discovery in Salmonella.

  1. It is not clear the method used for the classification of the sRNAs overlapped with coding sequences. Are these sRNAs oriented in an antisense manner with the genes? Or, as an alternative, are these RNAs produced by RNA decay (decRNAs) starting from a coding-protein mRNA? Please, clarify this point. I also suggest following another scheme of classification. For example, the ncRNAs that overlap with one gene and showing an antiparallel orientation are called asRNAs (antisense RNAs) or cis-RNAs, while the ncRNAs produced by RNA decay are called decRNAs. On the contrary, the ncRNAs coded from an intergenic sequence are called trans-RNA. I suggest renaming the classes of sRNA to improve the readability.

We largely agree with the reviewer’s suggestion that segregating reported sRNAs would somewhat improve readability and could potentially prove informative. That said, we have purposefully refrained from classifying sRNAs into existing annotations as we feel thorough evaluation of each individual sRNA is beyond the scope of this initial manuscript and the mechanisms driving individual sRNA expressions and their functional roles are better left to more focused evaluations. What’s more, we suggest that to classify a sRNA as a decRNA just because it overlaps a protein coding gene in the same orientation fails to consider other plausible alternative expression mechanisms (e.g. independent promoters). Similarly, we suggest many “trans-RNAs” might actually be embedded in unknown genes or extended UTRs from nearby neighboring loci. In addition, classifying a sRNA as an antisense RNA implies it is charged with regulating the gene on the opposite strand. It is conceivable, however, that a sRNA located opposite to a protein coding gene has no role in its regulation and may function entirely independently.

To better address this point in the text, the following has been added to the first paragraph of the discussion in the revised manuscript:

“That said, we have purposefully refrained from classifying sRNAs into existing annotations (e.g. asRNA, cis-RNAs, decRNAs, trans-RNAs) [20] as we feel thorough evaluation of each individual sRNA is beyond the scope of this initial manuscript, and defining the mechanisms driving individual sRNA expressions and their functional roles are better left to more focused evaluations.”

  1. In the genetic maps of figures, the strand of transcription of genes and sRNAs are not clear, so it is difficult to understand the correct orientation of the transcriptional unit. I suggest adding arrows representing the direction of the transcription and the tags 3’ or 5’ for the extremities (figures 1, 3, and 5).

As suggested, Figures 1, 3, and 5 and their legends have been modified to indicate transcriptional direction.

  1. What is the name of 4th line of Figure 2 panel B? Is it the marker? Please indicate it (in figure or/and in the caption). I also suggest indicating the MW near the arrow and, with another arrow, the MW of the sRNA.

We agree inclusion of a better description of the ladder facilitates easier interpretation by the reader and have updated the image and legend as suggested.

  1. The authors used IntaRNA and BLAST to predict the function of intergenic sRNAs, but it is not clear where the results are reported. Add these details in the caption of supplemental files.

Firstly, we thank the reviewer for noting this oversight. We apologize for mistakenly failing to include the full Supplemental Figure and Table Legends as a separate Supplemental File as was intended. This has been corrected.

In addition, the following text was added under target gene prediction in methods:

“Potential targets for each novel sRNA (Supplemental Tables 2, 3) were identified by both IntaRNA (rna.informatik.uni-freiburg.de) and alignment to current SL1344 genes annotated by Ensembl using BLAST+ (2.2.27) as previously reported [32].”

And the following was also added under Results 2.5. Identification of putative sRNA targets

“Strongly supporting the functional relevance of the our sRNAs, we find ~54% of our putative sRNAs either overlap the 5’ or 3’ ends of neighboring coding DNA sequences (CDS): mRNA, rRNA or tRNA loci (Supplemental Table 3); or share significant sequence complementarities to known mRNAs transcribed elsewhere in the SL1344 genome (Supplemental Table 2) [24] suggesting the principle function of the majority of the sRNAs identified in this work is to regulate transcripts via antisense base-pairing (Supplemental Tables 2,3).”

  1. The names of the Bacteria should be written in italics and the names of the species can be with the first lowercase letter. Please check all the text for this issue.

We thank the reviewer for catching this oversight and it has been corrected.

Reviewer 3 Report

The paper and topic is relevant to the scope of Antibiotics research.

The authors report upon their work reanalyzing NGS RNAseq read data from 2018 collected from Salmonella enterica serovar typhimurium exposed to Carbon-starvation. Small RNAs identified in the study were realigned to the S. enterica genome and sRNAs greater than or equal to 29 bp (as compared to 50 bp in the prior analysis) were characterized leading to the characterization of 475 novel and unannotated sRNAs. They conducted additional experiments including heat stress, hydrogen peroxide challenges, and Polymyxin B exposure with selected deletion mutants. Their Congo Red experiment demonstrated that one of the sRNAs may be involved in biofilm formation. The paper is well-written, the scientists conducted the appropriate experiments and the data is presented well.

Specific Comments:

  1. Salmonella enterica genus and species should be italics throughout the text.
  2. The blot in Figure 2 is missing sizing labels.
  3. How does the data in Figure 6 compare to the previous analysis?
  4. Page 7, lines 6-7: “both heat and heat (55 °C).” Please clarify.

Author Response

Reviewer 3 - Comments and Suggestions for Authors

The paper and topic is relevant to the scope of Antibiotics research.

The authors report upon their work reanalyzing NGS RNAseq read data from 2018 collected from Salmonella enterica serovar typhimurium exposed to Carbon-starvation. Small RNAs identified in the study were realigned to the S. enterica genome and sRNAs greater than or equal to 29 bp (as compared to 50 bp in the prior analysis) were characterized leading to the characterization of 475 novel and unannotated sRNAs. They conducted additional experiments including heat stress, hydrogen peroxide challenges, and Polymyxin B exposure with selected deletion mutants. Their Congo Red experiment demonstrated that one of the sRNAs may be involved in biofilm formation. The paper is well-written, the scientists conducted the appropriate experiments and the data is presented well.

We would first like to thank the reviewer for their thorough and largely positive evaluation of our work.

Specific Comments:

  1. Salmonella enterica genus and species should be italics throughout the text.

We thank the reviewer for catching this oversight, and it has been corrected.

  1. The blot in Figure 2 is missing sizing labels.

We agree inclusion of a better description of the ladder facilitates easier interpretation by the reader and have updated the image and legend as suggested.

  1. How does the data in Figure 6 compare to the previous analysis?

We agree this point warrants further discussion and therefore added the following to the Discussion:

As small colony variants (SCVs) and enhanced survivability are both potential indicators of increased persister cell formation [29], we suggest sRNA4130247 and/or sRNA4720054 may be involved in regulating the decision between active stress resistance and persister formation. What’s more, we find the sRNA transcriptome profiles of 5 and 24 hour starved cells to be highly distinct with few commonalities (Figure 6) and similarly observe highly distinct sRNA profiles associated with short duration (24 hr / recently fully desiccated) and prolonged (72 hr) desiccated cells [15]. Interestingly, however, we find significant overlaps between the sRNAs expressed during short duration desiccation and those expressed after 5 hours of C-starvation, and similarly find significant overlaps between the sRNAs expressed during prolonged desiccation and those expressed after longer 24hr C-starvation [11,15].  That said, as sRNAs are clearly vital to efficient cellular stress survival, sRNA participation in persister cell formation will undoubtedly be explored in the near future.

  1. Page 7, lines 6-7: “both heat and heat (55 °C).” Please clarify.

We again thank the reviewer for their thorough evaluation of our work. We believe, however, this issue arose from an easily understandable mishap during review as the only place we use “heat (55 °C).” in the article is on page 7 at lines 206 and 207 where the text reads:

Round 2

Reviewer 2 Report

The authors improved the manuscript according to the suggestion. I recommend accepting the manuscript in the present form.